# Feasibility Assessment of Wearable Respiratory Monitors for Ambulatory Inhalation Topography

**DOI:** 10.3390/ijerph18062990

**Published:** 2021-03-14

**Authors:** Shehan Jayasekera, Edward Hensel, Risa Robinson

**Affiliations:** Department of Mechanical Engineering, Rochester Institute of Technology, Rochester, NY 14623, USA; gbj6142@rit.edu (S.J.); echeme@rit.edu (E.H.)

**Keywords:** wearable respiratory monitors, smart garment, lung volume, respiratory inductance plethysmography, chest expansion, inhalation topography

## Abstract

*Background*: Natural environment inhalation topography provides useful information for toxicant exposure, risk assessment and cardiopulmonary performance. Commercially available wearable respiratory monitors (WRMs), which are currently used to measure a variety of physiological parameters such as heart rate and breathing frequency, can be leveraged to obtain inhalation topography, yet little work has been done. This paper assesses the feasibility of adapting these WRMs for measuring inhalation topography. *Methods*: Commercially available WRMs were compiled and assessed for the ability to report chest motion, data analysis software features, ambulatory observation capabilities, participant acceptability, purchasing constraints and affordability. *Results*: The following WRMs were found: LifeShirt, Equivital EQ02 LifeMonitor, Smartex WWS, Hexoskin Smart Garment, Zephyr BioHarness, Nox T3&A1, BioRadio, SleepSense Inductance Band, and ezRIP & zRIP Durabelt. None of the WRMs satisfied all six assessment criteria in a manner enabling them to be used for inhalation topography without modification and development. *Conclusions*: The results indicate that there are WRMs with core technologies and characteristics that can be built upon for ambulatory inhalation topography measurement in the NE.

## 1. Introduction

Historically, the technique for measuring chest motion was developed for investigating sleep quality [1] and in anesthesiology [2]. Chest motion was captured using wearable respiratory monitors (WRMs), which were comprised of sensors embedded in chest belts. More recently, WRMs comprise of sensorized garments which can be worn as undergarments or shirts. Many WRMs are still marketed for sleep studies and are standard in polysomnography (PSG) equipment. However, some of the WRMs have sensors that measure physiological parameters beyond just those related to respiration, such as heart rate (HR) and heart rate variability (HRV), as well as accelerometers for tracking activity (e.g., walking, standing, sleeping, etc.) and for measuring posture and position. As a result of these new multi-parameter WRMs, the potential use cases have expanded to include many application domains. Researchers have used these multi-parameter WRM systems in military applications [3] to track operational performance and stress levels in soldiers, in fitness and sports [4], and in homecare of diseased or elderly patients [5]. In these applications, the metrics used are typically HR, HRV, and respiratory rate (RR). However, there are other applications that can benefit from the raw, unprocessed signals of chest motion reported by the WRMs.

Tobacco regulatory sciences is one domain that can benefit from the raw respiratory signals obtained from WRMs. In particular, WRMs may be capable of measuring inhalation topography associated with tobacco use [6,7]. Inhalation topography (i.e., volume, duration, flow rate, start and end times of inhalation and exhalation, breath-hold and time between breathing cycles) provides information that is required for accurately modeling lung deposition of inhaled particles [8], including tobacco product emissions and inhaled medications. For example, longer exhalation times are associated with increased deposition in smokers [9]. Deeper, more intense, inhalations are associated with compensatory behavior in switching to low yield products [10].

Studies have since found there are a number of possible compensation mechanisms to adapt to low nicotine products, [11,12], including altering the manner in which they puff (puffing topography) [13,14,15] altering the manner in which they inhale and exhale the puff (inhalation topography) [16,17,18,19,20] and increasing their overall frequency of use [21], [22,23]. Although techniques are available to assess puff topography [24,25,26,27,28] and overall frequency of use in the natural environment (NE), little is known about inhalation topography in the NE. Commercially available WRMs make it possible to explore ambulatory measurement of various physiological parameters, yet little work has been done to exploit WRM technology for the purpose of measuring inhalation topography.

Konno and Mead [29] presented a theory relating chest motion to lung volume using two signals; thoracic (TC) and abdomen (AB). Since most WRMs operate on the basis of measuring chest motion, there is the potential to use the WRMs to convert the chest motion signal into a lung volume waveform (lung volume as a function of time) and then use that lung volume waveform to derive the desired inhalation topography parameters. One method to derive these is by locating the start and end times of the inhalation and exhalation portions of each breath in the volume waveform. Once these time points are determined, the inhalation and exhalation volumes and durations can be derived. Inhalation volume is the rise in volume during the inhale period and exhalation volume is the drop in volume during the exhale period. Inhalation duration is the period between the start of the inhale and the end of the inhale, and likewise the exhalation duration is the period between the start of an exhale and the end of an exhale. The breath-hold period is the duration between the end of an inhale and the start of an exhale. The cycle duration is the period between the start of an inhale and the end of an exhale. The mean flow rate is calculated by dividing the volume by the duration. Preliminary data demonstrating the use of WRMs to obtain lung volume was presented [6,7], but these studies only covered a few products, and did not demonstrate ambulatory measurements in the NE. More work is needed to better understand the technology landscape related to ambulatory measurement of lung volume in the NE.

This paper provides a review of commercially available WRMs with the purpose of assessing the extent to which each may be suitable for obtaining inhalation topography in the NE. With additional engineering developments, WRM can assist in differential risk assessment of new and emerging tobacco products.

## 2. Overview of WRMs

A WRM is a system comprising of a sensorized garment, a datalogger, and data analysis software. The garment is worn on a person’s torso with the capability of measuring the TC or AB (or both) motions using embedded strain gauge, piezoresistor, or inductance sensors. Commercially available WRM garments come in different form-factors: belts, straps, and shirts. (a) Belt—individual adjustable length belts made from fabric with embedded sensors typically worn above clothing but may also be worn below clothing, directly on the skin. When worn over clothing, they can be tightened to get the necessary fitness to achieve accurate chest motion measurements. Each belt can be placed at the desired location to measure either TC or AB motions. With two belts, both TC and AB can be measured simultaneously; (b) Strap—made from elastic form-fitting material, with one or two shoulder straps. There are also some without any shoulder straps, making them similar to the belts but the key distinction is in their elasticity and how they are worn. Straps are worn as undergarments, on the upper torso, directly on the skin. Due to their placement, straps only measure TC motion; (c) Shirt—made of tight form-fitting material or slightly looser breathable material, worn either as an undergarment or as a standalone shirt (not requiring a shirt be worn over it). Similar to the strap type garments, shirts are elastic and require skin contact for a tight fit, but the key distinction is that the shirts cover the full torso and may be worn in place of regular clothing. Shirts typically have embedded sensors that measure both TC and AB or TC only. Figure 1 shows exemplar WRMs of each type.

The embedded sensors in the garment are powered using an ancillary device which typically doubles as the signal conditioner (e.g., analog to digital conversion, signal noise filtering, etc.) and data collection device. In this article, these devices will be referred to as a datalogger. In addition to storing data locally on the device, some dataloggers have the ability to transfer data wirelessly in real-time to a mobile or computer device. Dataloggers in multi-parameter WRMs may also measure and store data from other non-chest motion related sensors that are available on that system such as signals from ECG probes, skin temperature from thermocouples, blood oxygen saturation from pulse oximetry, etc.

WRMs typically come with software to offload and organize data collected on the datalogger. In some WRMs, this software is also used to calculate and report parameters such as HR, HRV, and RR. Some WRMs come with software that aids in calibrating the monitors to convert the raw chest motion waveform to lung volume waveform.

## 3. Methods

Commercially available WRMs were reviewed and assessed in four steps: (1) Information about commercially available WRMs was gathered, (2) assessment attributes important to measuring inhalation topography in the NE were defined, (3) the product characteristics associated with each assessment attribute were defined, and (4) a scoring rubric to evaluate the WRMs was defined and used.

### 3.1. Gather Information about Commercially Available Wrms

Online searches were conducted to find commercially available WRMs that performed chest motion measurements as part of its operation. Product characteristics were gathered from vendor websites, product brochures, and specification sheets. In cases where available information was lacking, sales associates, technicians, and engineers were contacted. Product characteristics included type of sensor technology used, placement of the sensors (i.e., TC and/or AB), and sampling rate. Information on battery lifetime and storage capacity, garment style, how the garment attached to the datalogger and the amount of interaction required by the participant to operate the monitor were also collected. In addition, streaming capabilities, secondary sensors, software compatibility and cost were recorded. Any potential benefits and limitations were noted, such as whether the vendors make the raw chest motion data available and whether the WRM has been used by others to derive lung volume from raw chest motion, along with the vendor’s advertised application for each WRM, and literature on studies that used each WRM.

### 3.2. Define Assessment Attributes Important to Measuring Inhalation Topogography in the Natural Environment

We defined six key assessment attributes related to conducting studies of inhalation topography in the NE: (I) Chest Motion Data, (II) Data Analysis Software, (III) Ambulatory Observation Capabilities, (IV) Participant Acceptability, (V) Purchasing Constraints, and (VI) Affordability. Chest Motion Data assesses the ability of each WRM to capture chest motion, including the type, number, and location of the sensor used. Data Analysis Software assesses the bundled analysis software of each WRM for data pre-processing, calibration, and calculating inhalation topography. Ambulatory Observation Capabilities assesses the ability to deploy the WRM in the NE with minimal study coordinator and participant intervention. Participant Acceptability assesses the likelihood that a study participant will be willing to use the WRM for an extended period of time as they go about their day-to-day activities. Purchasing Constraints assesses the availability, purchasing restrictions, and ease of purchasing of each WRM. Affordability assesses the overall cost of acquiring and deploying each WRM. Details of each assessment attribute are presented in the next section along with WRM product characteristics associated with each attribute.

### 3.3. Identify the Product Characteristics Associated with Each Assessment Attribute

We identified the product characteristics (PCs) associated with each of the six assessment attributes. These characteristics were those considered to be important to affecting the attribute and, unless otherwise noted, these were all used in the subsequent assessment rubric.

#### 3.3.1. PCs Related to Chest Belt Motion

In order to achieve the goal of assessing inhalation topography, the WRM’s datalogger must record raw chest motion waveform (chest motion as a function of time). Only WRMs that record the raw chest motion waveform are appropriate for inhalation topography measurement. Any WRMs that do not record chest motion waveform will not be acceptable for further evaluation.

An important consideration is the number of chest motion sensors and their locations. The model by Konno and Mead requires two chest motion signals; one at TC and another at AB. While a WRM which only records either TC or AB chest motion data may be acceptable for measuring RR, WRMs which record both TC and AB motions are preferred. In particular when the relative motion of TC and AB are of interest. For example, some smokers may inhale deeper by expanding their abdomen instead of their thorax as a mechanism of compensatory behavior. If only TC is measured, this behavior would be missed. The importance of relative motion of TC and AB is well established in sleep studies, in particular in the diagnosis of sleep apnea.

In addition to the number and location of sensors, the sampling frequency at which the data is recorded ultimately impacts the fidelity of the resulting inhalation topography. The sampling frequency directly affects the accuracy of time-based inhalation topography parameters such as inhalation and exhalation durations, which in turn would affect the accuracy of the inhalation and exhalation flow rates. The appropriate sampling frequency would depend on the desired accuracy of these measured parameters. Given a typical breathing cycle of around 4 seconds, split evenly between inhalation and exhalation, to obtain an accuracy of 5% of the inhalation or exhalation duration, the sampling period must be at least 0.05 s (2.5% of 2 s to satisfy Nyquist requirement) or 20 Hz. A sampling rate of around 5 times this would be necessary to see and characterize the transient features of the inhalation and exhalation waveform. Additionally, this sampling frequency or higher would also help identify breath-hold, which is typically a fraction of the inhalation or exhalation durations. For our application, we also intend to synchronize the lung volume waveform from a WRM with the flow rate waveform from our puffing topography monitor to associate inhalation topography and puffing topography. We established that a sampling rate less than or equal to 100 Hz would provide marginal fidelity while a sampling rate greater than 100 Hz is likely to provide sufficient fidelity for our needs.

The type of TC or AB sensor could also influence the signal response. Sensor technologies employed in commercially available WRMs include strain gauge, piezoresistor, capacitive sensor, and respiratory inductance plethysmography (RIP). No study has been found that has investigated the impact of the sensor type on ambulatory lung volume measurement. The performance specifications including accuracy, range, precision, stability, and repeatability of the sensors are important but are not reported by manufactures. Since there is no information on the impact of sensor type and on the performance specifications of each WRM, these are not included in the assessment rubric. One factor that may impact the sensitivity of a particular sensor type is the effective range of motion that it captures. For example, the RIP sensor measures the changes in the enclosed cross-sectional area within the band of the sensor whereas the other sensor types measure the change in circumference of the band as measured at the point of sensor placement. Qualitatively, the RIP sensor may therefore be more sensitive to changes in the chest wall motion than the other sensor types.

#### 3.3.2. PCs Related to Data Analysis Software

Some vendors may provide software which processes the raw chest motion data described in assessment attribute (I). The chest motion signals measured by the WRM sensors need to be conditioned before they can be used for calculating lung volume. This conditioning may include filtering noise and motion artifacts, and compensating for sensor drift. Some WRM dataloggers perform signal conditioning internally as the data is being collected, which allows them to calculate comprehensive metrics such as HR and RR. However, these dataloggers are typically “black boxes” and users may not have access to their proprietary algorithms. Other WRM dataloggers do not perform signal conditioning on the raw chest motion data; users would need to develop and apply their own preprocessing software.

After signal conditioning, the chest motion signals can be calibrated to known lung volume measured using a spirometer or other primary instrument. Some WRM manufacturers may provide software for calibrating conditioned chest motion data to lung volume data. If the calibration software is not provided, the user must develop their own algorithm or rely on third-party software. The correlation between the signal response from the sensor of the WRM to chest motion is a fundamentally important factor for calculating lung volume. There must be a one-to-one relationship between sensor response and chest motion for calibration to be successful. Currently, the only known commercially available software for this purpose is VivoSense (VivoSense Inc., Newport Coast, CA, USA). Academic research labs have developed software, such as TAP™ [30], to accomplish this calibration.

Some WRM manufacturers provide a software development kit (SDK) or an application programming interface (API) to interact with their WRMs. The SDK/API may provide access to either the raw chest motion data, the conditioned chest motion data, or both. Availability of an SDK/API would benefit those looking to develop their own signal conditioning, calibration, and/or inhalation topography analysis software.

Once the sensor signals have been conditioned and the chest motion waveform calibrated to a known volume, the lung volume waveform is obtained. The fidelity of that volume signal is limited by the sampling rate which was included in assessment attribute (I). Analysis software is required to determine the breathing cycle parameters for the inhalation topography parameters, including breath start time, inhalation flow rate, volume and duration, breath hold duration, exhalation flow rate, volume and duration, and pause between breaths. Currently, there is no known commercially available software for this purpose. Since inhalation topography requires identifying the start and end of each inhale and exhale, this process is not trivial, especially when dealing with long observations with many breathing cycles. A WRM that comes bundled with software that provides inhalation topography would directly suit our needs whereas a WRM that comes with no software would require significant engineering work to make it acceptable.

#### 3.3.3. PCs Related to Ambulatory Observation Capabilities

In order to achieve the ultimate goal of assessing ambulatory inhalation topography in the NE, the WRM must be carried on a person’s body and not be tethered to a stationary device, such as a desktop computer or an AC power source. Only WRMs meeting this constraint are appropriate for ambulatory inhalation topography measurement. The untethered constraint suggests that the product characteristics of data storage capacity and battery life are paramount.

To assess this attribute, we considered three categories of observation period. The minimally acceptable observation period would consist of ambulatory measurements over a short duration of time less than or equal to 8 h. The next level of observation period performance would support ambulatory measurements over a single 24-h period, while the third and preferred level of performance would support ambulatory observation over multiple days.

For multi-day observations, the participant might have to charge the WRMs or off-load data themselves, or frequently return to the lab. The implication of requiring a study participant to periodically recharge the datalogger was viewed less adversely than frequent downloading of data from the datalogger or frequent lab visits. The requirement to frequently charge the datalogger and download data puts a burden on the participant and may potentially reduce compliance and increase risk of data loss.

#### 3.3.4. PCs Related to Participant Acceptability

In order to achieve the ultimate goal of assessing ambulatory inhalation topography in the NE, participants using the WRM must be able to put the garment and datalogger on by themselves. WRMs that do not meet this constraint are not acceptable for further evaluation. For ambulatory observation of inhalation topography in the NE, there are many factors that influence participant acceptability, including how well the garment fits and feels against the skin, how easy it is to wear and remove, and how it interfaces with and affects the participant’s own clothing. High acceptability leads to better compliance and reduces risk of data loss.

The way a WRM garment is to be worn (underneath or over regular clothing, or worn on its own) can greatly influence its acceptability. Wearing a WRM above the participant’s own garment may make the monitor stand out in a negative way. Monitors that can be worn under regular clothing are more discreet and therefore more likely to be acceptable. Cumbersome WRMs and WRMs with wires that are clearly visible, or dataloggers that must be carried separate from the garment are more likely to reduce participant acceptability. A discreet and non-cumbersome WRM would not distract the participant from the task they are performing and would allow the participant to feel more at ease using it, especially if the observation is conducted in a public setting. Since the WRMs considered here are commercial systems, it is expected that the manufacturers have considered the ergonomics and aesthetics when designing them. However, the proposed use of the WRM for ambulatory lung volume measurement in the NE may differ from the use-case scenarios they were originally designed for.

#### 3.3.5. PCs Related to Purchasing Constraints

The ease with which a WRM can be obtained is of importance for timely research. WRMs with limited availability or long lead times may not be suitable for some researchers. Although information may be present online and in the literature, some WRMs may not be available because they are under development or housed in an academic lab. Some WRMs may be difficult to purchase because the vendor only sells to clinicians or healthcare providers.

#### 3.3.6. PCs Related to Affordability

The cost of a WRM has an impact on the budget considerations of a study. There are various factors that affect the overall implementation cost of a WRM. The cost of each subcomponent in a WRM that can be purchased separately. In some WRMs, the dataloggers and garments are sold separately, with the former typically more expensive than the latter. For a large study group, it may be more suitable then to invest more on the garment and have them in as many sizes as possible while having fewer dataloggers. This will allow the researcher to accommodate a varied study group in terms of body sizes. This is a trade-off between number of concurrent observations and cost. Some WRMs come in different styles for men and women. This can potentially increase the cost of the study since male and female garments of each size would need to be purchased. Most WRMs are washable and reusable but some WRMs, in particular the chest belt type monitors, are disposable. The disposable garments are typically cheaper but could cost more over time. Some WRM garments come in many narrow size ranges, thus requiring a larger number of wearable components to encompass the entire range of sizes. Whereas some WRMs, such as the chest belt type, are adjustable and thus a single unit may be able to fit most body sizes.

### 3.4. Define and Use Scoring Rubric to Evaluate the WRMs

The six assessment attributes were evaluated by rating the product characteristics on a linear scale from 1 to 5. Level 1 indicates there is no evidence the WRM will meet the needs for ambulatory inhalation measurement. Level 2 means the WRM may meet the need with significant engineering effort. Level 3 means the WRM currently meets the need but in a suboptimal way. Level 4 means the WRM currently meets the need but has not been validated. Level 5 means the WRM meets the need and has been validated by a third party in a peer reviewed publication. Levels 1 through 5 were interpreted for the product characteristics associated with of the six assessment attributes. The final scoring rubrics are shown in Table 1.

## 4. Results

### 4.1. List of WRMs and Product Charactersitics

A total of 9 WRMs were identified. Below is a brief description of each, highlighting the product characteristics with attention to their suitability for use in NE to capture ambulatory lung volume. The WRMs described below are grouped by type: WRMs 1–7 are multi-parameter systems with sensor, datalogger, and software packaged together, WRMs 8–9 are chest belts comprising of only the respiratory sensors (i.e., garment only) and require custom made or third-party datalogger and software. Details of WRMs 1–7 are compiled in Table 2.

#### 4.1.1. LifeShirt

The LifeShirt (Vivometrics Inc., Ventura, CA, USA) was one of the first non-invasive multi-parameter systems for personal health monitoring. It was novel in that it was able to measure cardiovascular parameters, including HR and HRV, and pulmonary parameters, including RR and minute ventilation volume (V_E_), non-invasively, for extended periods of observation, within free-living environments. The system included ECG leads, a thoracic RIP band, an abdominal RIP band, a motion detector, and an electronic diary. The electronic diary was used for gathering subjective patient data that could be programmed to ask questions relevant to the research being done. All the data was securely stored on a flash memory card, which could then be later accessed by clinicians or researchers. An analysis software called VivoLogic (the precursor to VivoSense) developed by the vendor was able to analyze the data and derive a multitude of parameters. At the time of this writing, the LifeShirt is no longer available for purchase.

The system was tested and featured in numerous studies with applications ranging from sleep studies to automobile racing to U.S. Air Force pilot-testing. The system has been reviewed by these authors [31,32]. The validity of cardiopulmonary measurements by the LifeShirt were assessed during exercise [33] and in obese patients [34], both conducted in a laboratory setting. It was used to measure lung volume during exercise in both healthy and unhealthy patients [35], measure lung volume in patients with severe pulmonary disorders [36,37,38], measure respiratory parameters related to induced stress [39], measure end-expiratory lung volume [40], and measure V_E_ [41]. These were conducted either in a laboratory or clinical setting, with the exception of [37] which was conducted in both a laboratory and a home environment.

#### 4.1.2. Equivital EQ02 LifeMonitor

The Equivital (Hidalgo Inc., Swavesey, UK) EQ02 LifeMonitor is a chest-strap type wearable monitor equipped with ECG probes, a skin thermometer, a 3-axis accelerometer, and a strain gauge chest motion sensor. The LifeMonitor records all signals continuously and stores the data on a detachable datalogger module, called the Sensor Electronic Module (SEM). The SEM is also able to stream the signals to a mobile device or computer via Bluetooth (additional software must be purchased). LifeMonitor settings can be changed and data can be extracted via an accompanied software. The chest motion sensor is located at the thorax and is used to report RR. This, along with HR and HRV from the ECG probes, are calculated continuously using raw data from the previous 15 s. It is also able to infer body position and posture based on the accelerometer data. The LifeMonitor seems to be geared towards clinical research and personal fitness applications. The LifeMonitor comes in multiple unisex sizes, with fit-adjustment clasps, and is washable.

No studies have been found that reported using the LifeMonitor to measure lung volume or inhalation topography. A number of studies [42,43] used the LifeMonitor to track qualitative respiratory parameters. The LifeMonitor’s viability for use in military applications were assessed in a number of studies [3,44,45]. A study by [46] aimed to validate the LifeMonitor; they compared the performance and reliability of the LifeMonitor against standard calibrated laboratory physiological monitoring devices on adult male participants (N = 6) under “resting, low, and moderate-intensity” activities for heart rate, respiratory rate, multipoint skin temperature, and core temperature. The results indicated high correlation between the LifeMonitor and the standard laboratory devices for all physiological measures and they concluded that the LifeMonitor is suitable for multi-parametric ambulatory health monitoring.

#### 4.1.3. Smartex WWS

The Smartex (Smartex s.r.l., Pisa area, Italy) Wearable Wellness System (WWS) is a shirt type multi-parameter WRM comprised of a sensorized garment and a data capture module (called SEW) which can fit in the garment. The Smartex WWS garment includes ECG probes and a piezoresistive sensor for measuring chest motion at the thorax. In addition to logging data and supplying power to the sensors embedded in the garment, the SEW is also able to capture body movement using a built-in 3-axis accelerometer. Using the raw ECG, respiratory, and body movement signals, the Smartex WWS is able to report HR, HRV, RR, posture, step count, and estimated energy expenditure. The Smartex WWS can also stream the physiological signals live to a computer via Bluetooth. It is worn as an undergarment and is washable.

No studies were found that reported using the Smartex WWS to measure lung volume or inhalation topography. A number of studies used it for measuring qualitative physiological parameters in studying sleep disorders [47,48] and in Alzheimer’s rehabilitation [49]. A general trend that was observed in the literature is that the Smartex WWS was often used as part of a larger system of instruments and devices for the purposes of monitoring the elderly, with focus on rehabilitation and disease identification.

#### 4.1.4. Hexoskin Smart Garment

The Hexoskin Smart Garment (Carré Technologies Inc., Montréal, PQ, Canada) is a shirt type multi-parameter WRM comprised of a sensorized undergarment and datalogger. The Hexoskin garment contains embedded ECG leads, RIP respiratory sensors at both the thorax and the abdomen, and a 3-axis accelerometer. The Hexoskin reports HR, HRV, RR, V_E_ (reported in arbitrary units), body position, and step count. The datalogger (called the Hexoskin Smart Device) stores all the raw signals and is able to stream these signals to a computer or smartphone via Bluetooth. The Hexoskin garment is washable and is made of form-fitting, lightweight, breathable fabric.

No studies were found that reported using the Hexoskin to measure lung volume or inhalation topography. It was used as a reference instrument in comparisons with other wearable monitors with mixed results. A comparison with a novel system called DoppleSleep showed similar results [50] for HR and RR. Another study showed high correlation on RR but poor correlation in HR when the Hexoskin was compared to a Polar T-31 heart rate monitor and an Applied Electrochemistry Moxus Metabolic System [51]. The validity of the Hexoskin’s measurement of HR, RR, and V_E_ were assessed in a number of studies [52,53], with results indicating that the HR and RR showed strong correlation to the reference standard.

#### 4.1.5. Zephyr BioHarness

The Zephyr BioHarness (Zephyr Technology Corp., Annapolis, MD, USA) is a chest-strap type multi-parameter WRM. The garment has embedded sensors that measure ECG, and respiratory signals by measuring motion of the thorax. It also has a 3-axis accelerometer and thermistor. The BioHarness reports HR, HRV, RR, activity level, posture, and skin temperature. It comes with a datalogging module that attaches directly to the garment and provides power to the sensors. It is able to stream data wirelessly via Bluetooth. The BioHarness garment is washable.

No studies were found that reported using the BioHarness to measure lung volume or inhalation topography. The BioHarness was validated for RR [54], used for measuring RR [55], and used for measuring HR and HRV [56]. The BioHarness was used in a study [57] to generate physiological data for aiding in the development of an algorithm for analyzing mixed sampling rate time series data. It was also integrated into a wearable system for monitoring physiological parameters of fire fighters [58].

#### 4.1.6. Nox T3 and Nox A1

The Nox T3 and the Nox A1 (Nox Medical Global, Reykjavík, Iceland) are multi-parameter portable PSG devices that interface with the vendor’s external sensors including RIP belts, nasal cannula, and pulse oximeter. The A1 can also interface with electroencephalogram (EEG) probes. The vendor supplies chest belts with embedded RIP sensors that once attached to the Nox, allows it to measure TC and AB waveforms. The belts are typically for single use. The Nox has an integrated microphone to detect snoring and a 3-axis accelerometer for measuring body motion and position. The Nox device has Bluetooth capabilities. It is marketed for at-home sleep testing.

One study [7] reported using the Nox T3 for measuring lung volume during the 6-minute walk test for evaluating exercise capacity. The data from the chest motion was analyzed using the group’s own custom developed algorithms. The study reported differences in tidal volume, and other parameters, across the two cohorts tested but did not report actual lung volumes.

#### 4.1.7. BioRadio

The BioRadio (Great Lakes NeuroTechnologies, Independence, OH, USA) is a multi-parameter portable PSG device with the ability to record many physiological signals including EEG, electromyogram (EMG), electrooculography (EOG), and electrocardiogram (ECG), and respiratory signals by attaching the corresponding sensors. It has an embedded 3-axis accelerometer. Similar to the Nox, the vendor supplies RIP chest belts that must be purchased separately and attached to measure lung volume at both the thorax and the abdomen. It can store data onboard for offline analysis or stream to a computer via Bluetooth. One study reported using the BioRadio with two RIP chest belt attachments for measuring tidal volume in infants [6]. The cohort mean tidal volume measurements were reported along with an exemplar tracing of the abdominal and the thoracic lung volumes. The BioRadio was also used to report RR in adults performing music [59].

#### 4.1.8. SleepSense Inductance Belt

The SleepSense Inductance Belt (S.L.P. Inc., Elgin, IL, USA) is a chest belt with an embedded RIP sensor. The belt could be worn either on the thorax or the abdomen. Both TC and AB can be measured if two belts were worn at the same time. The belts are easily adjustable to fit different body sizes. The belt is designed for in-laboratory lung volume measurements and is standard in PSG equipment. The belt may be adapted for ambulatory measurement with the use of a custom or third-party datalogger to collect and condition the data. The belt comes in both reusable/washable and disposable variants.

One research group [60,61] reported using the SleepSense for measuring lung volume. They used the SleepSense as part of a system (called PACT) developed by their laboratory for measuring cigarette smoking behavior. The SleepSense was also used as part of polysomnography in a few studies [62,63] and to evaluate a novel respiratory sensor [64].

#### 4.1.9. ezRIP & zRIP DuraBelt

The ezRIP and zRIP DuraBelt (Koninklijke Philips Electronics N.V., Amsterdam, Netherlands) are chest belts with embedded RIP sensors and functionally work the same way as the SleepSense Inductance Belts. Likewise, these are designed for in-lab lung volume measurement but may be adapted for ambulatory lung volume measurement with the use of a custom or third-party datalogger.

Researchers have used the DuraBelts as part of polysomnography equipment in sleep studies and for investigating sleep disorders [65,66]. Only one article, by the PACT group, reported using the zRIP as part of a system to measure lung volume in regular smokers [67]. The data were primarily used to develop an algorithm for data analysis and no actual lung volume results were reported.

### 4.2. WRM Assessmet Results

Figure 2 illustrates the results of the scoring on each assessment attribute as per the rubric described in Table 1. WRMs 8 and 9 were not included because they are not complete WRMs (missing datalogger and software). A radar chart is used since a simple cumulative score is insufficient to compare the various WRMs given their different strengths and weaknesses. The radar chart allows the WRMs to be compared holistically as well as on each assessment attribute separately.

## 5. Discussion

There are clear gaps in the state of the art when it comes to using commercial WRMs for ambulatory measurement of inhalation topography in the NE. One clear gap is in the analysis software. None of the WRMs reviewed here come with analysis software that provide inhalation topography (score of 4 or above). Only two WRMs, the LifeShirt and the Nox, come with analysis software that provides lung volume. The engineering difficulty of going from chest motion waveform to lung volume waveform (score of 2 to 3) is significantly lower than going from lung volume waveform to inhalation topography (score of 3 to 4). Many calibration techniques, a necessary step for going from chest motion waveform to lung volume waveform, have been developed, assessed, and reported over the years. In comparison, there have been very few reports of work being done to obtain inhalation topography from lung volume waveform [68,69]. The primary challenge is in identifying each breath in the volume waveform. This involves locating the start and end of an inhale and the start and end of an exhale for each individual breath. This is made difficult by the large amount of data involved; assuming normal tidal breathing of around 20 breaths per minute, that would yield over 30,000 breaths over the course of a 24-h observation period per participant. Additionally, since breaths occur sequentially, any error in identifying the end time of one breath could result in an error in the start time of the following breath. Furthermore, irregularities in the signal, either due to abnormal sensor response or motion artifacts, could further complicate the analysis process. These irregularities if not handled correctly could result in breaths being misclassified, i.e., spurious volume motion identified as a breath, which can impact the overall accuracy of the mean inhalation topography parameters. This is a challenging step that currently requires significant development to overcome for all WRMs.

Furthermore, in studying particle deposition from tobacco emissions in the lungs, it is necessary to identify and discriminate between tobacco-use breaths (i.e., vaping or smoking) and normal non-tobacco-use breaths. This capability is not present in the WRMs or the associated software. The process of automatically identifying tobacco-use breaths requires an external reference to establish the onset of tobacco-use behavior (i.e., start of a puff). Without this, it would not be possible to truly discriminate between normal breathing and smoke/vapor inhalation. One method that has been proposed in the literature is to use hand-to-mouth gestures [70] to establish the start and end of a puff. Another approach is to use RIT’s wPUM™ puff topography monitors [26], which report the time of day of each puff. The start time and end time of a puff will aid in identifying the associated breath in the volume waveform. This is an area that is currently not addressed by the current commercial WRMs.

Another clear gap is that no WRM received a score of 5 on attributes I to IV, indicating that no validated and published data have been found to support suitability of a commercial WRM for measuring ambulatory inhalation topography in the NE. Holistically the Hexoskin is the most suitable on account of its high score on most of the attributes. However, the Hexoskin, along with the other WRMs presented here are not immediately ready for ambulatory inhalation topography measurements in the NE, since none received score of 5 on every attribute. A score of 4 on attributes I to IV would indicate that the WRM is potentially ready, barring validated and published data. Most of the WRMs received a score of 4 in at least one of those attributes, which depending on the WRM and what attributes received a lower score, could either mean that some engineering work must be done before the WRM is ready or that there might be limitations to the study objectives or protocols if that WRM were used. It should be noted that on the radar chart, the LifeShirt appears to be the least suitable, but that is due to it receiving a score of 1 on many attributes, which is not due to some inherent weakness but because of lack of information.

There are two PCs in attribute I that were identified but not included in the assessment rubric. These are the chest motion sensor type and the performance specification of the sensor. The sensor type is disclosed by the manufacturer, but the performance specifications are not. It is possible that the different sensor technologies may have an impact on the WRMs ability to measure ambulatory inhalation topography but no research was found in the literature that robustly characterized each of these sensor types in the context of measuring these parameters. Some WRMs were not explicitly designed to measure lung volume and only use the chest motion sensor to report RR, which can be calculated even when there is poor correlation between breathing and chest motion.

One area that was not addressed in this paper is how well a WRM is able to be calibrated to obtain the volume-motion (V-M) parameters that relate TC and AB to the volume waveform, and how well it is to maintain this calibration throughout the observation period. Some WRMs may be more resilient than others to changes that result in a shift in V-M parameters. Product characteristics such as sensor quality, type, and placement location may influence the WRM’s ability to hold calibration. In addition to product characteristics, the accuracy and stability of calibration is also dependent on the participant’s physiology and breathing style, as well as the conditions and activity being observed. This assessment, therefore, cannot be made without conducting a robust study on human subjects with these WRMs.

The performance specifications of the sensor limit the accuracy of the subsequent inhalation topography results derived from measurements made on that sensor. As such, the viability of a WRM for measuring ambulatory inhalation topography is dependent on the performance specifications of the WRM’s sensors. Unfortunately, this information is not disclosed by the manufacturers. At best, the manufacturers provide a subset of the performance specifications for the calculated metrics such as HR, HRV, and RR. Once again, no research was found in the literature that robustly characterized the performance specifications of the sensor on these WRMs.

The assessment of participant acceptability is the most subjective of all the assessment attributes. It is not possible to robustly assess this attribute without getting feedback from a focus group sampled from the target demographic. Some WRMs, such as the Hexoskin and Zephyr BioHarness, are designed for extended and active operation. The belt type monitors, such as the SleepSense, ezRIP/zRIP, Nox T3/A1, and BioRadio, are mainly designed for sleep studies and may not be comfortable for other activities. Furthermore, these belt type monitors do not come as an integrated garment, like the Hexoskin, Equivital EQ02, Smaretex WWS, and the Zephyr BioHarness, where the datalogger is ergonomically and discreetly attached to the garment. For this reason, the belt type WRMs would tend to be cumbersome and indiscreet once all the components are attached, whereas the shirt type WRMs would tend to be sleeker and more discreet.

It is important to note that the scores (Figure 2) given to each WRM on each assessment attribute reflects the opinions of the authors. These assessments were made based on the information available to the authors and tabulated on Table 2. We do not assert that this evaluation is absolute and in fact we invite researchers to evaluate each WRM and interpret the results in the context of their own research goals and constraints. We present the assessment rubric (Table 1) in hopes that it would serve as a good framework or a starting point for doing such evaluations.

WRM technology has evolved over the years, starting from simple belts (such as the SleepSense Inductance Belt) to straps (such as the Equivital EQ02 LifeMonitor) and to now fully integrated shirts (such as the Hexoskin Smart Garment). With each evolution comes new features and functionality that expand their usability. Unlike belt-type WRMs that are bulky and only measure chest motion, making them only suitable for limited mobility applications like sleep studies, strap-type WRMs are easier worn, essentially acting as an undergarment, which reduces its impact on mobility. This made them suitable for sports and fitness applications, which also benefited from the various embedded sensors, such as the heart rate monitor, accelerometer, and thermocouple to measure skin temperature. The downside to the strap-type WRMs is that they only measured chest motion at the thorax. Recently, the focus has been to develop a garment with a suite of embedded sensors for a broad range of applications. The shirt-type WRMs improve upon the strap-type by potentially eliminating the need for an additional garment and having space for more than one respiratory sensor. In the future, we anticipate WRMs to continue the trend of embedding more sensors, such as an electromyograph to measure muscle activity, with the introduction of Internet-of-Things, WRMs may also interface with other body sensors to form a body area network. These new advancements would enable a wider application in many domains. In tobacco research, one benefit would be in enabling simultaneous observation of smoking/vaping behavior and inhalation behavior by linking a wireless puffing topography monitor to a WRM. This would eliminate the current difficulty in needing to synchronize data from the two sources.

## 6. Conclusions

A total of nine commercial WRMs were identified and briefly reviewed. The technological readiness of each WRM was assessed and reported. None of the nine products completely meets the needs of measuring ambulatory inhalation topography in the NE. Three very clear gaps have emerged as a result of the assessment: (1) No WRM comes equipped with analysis software to derive inhalation topography from lung volume waveform; (2) No work has been found that has validated any of the WRMs for ambulatory inhalation topography measurements; (3) The measurement accuracy of inhalation topography as a function of the WRM’s sensor type or sensor performance specifications have not been found. Further development is needed to fill all three gaps in order to conduct a successful human-subject clinical trial. Overall, the results indicate that there are WRMs with core technologies and characteristics that can be built upon for ambulatory inhalation topography measurement in the NE.

## Figures and Tables

**Figure 1 ijerph-18-02990-f001:**
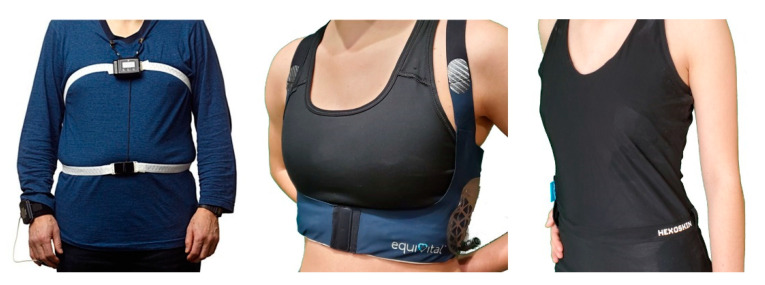
Exemplar WRMs of each type: belt (**left**), strap (**center**), and shirt (**right**). The belt type WRM shown is the Nox T3 with two RIP bands (TC and AB) worn above clothing. The strap type WRM shown is the Equivital EQ02 LifeMonitor, worn alongside a fitness undergarment. The shirt type WRM shown is the Hexoskin Smart Garment, worn directly on the skin. Photo credits: (**left**) image reproduced with permission of Nox Medical Global, Reykjavík, Iceland; (**center**) and (**right**) images taken by Respiratory Technologies Laboratory.

**Figure 2 ijerph-18-02990-f002:**
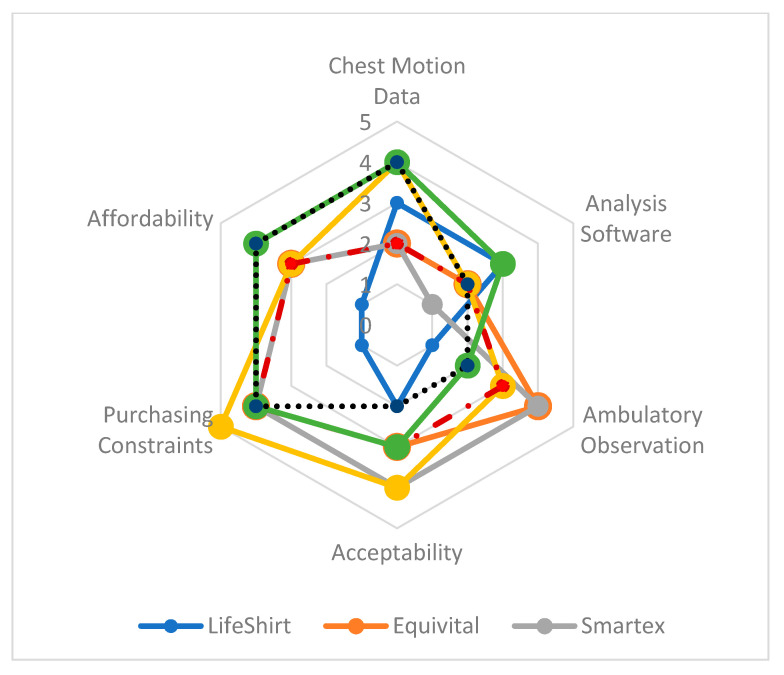
WRM assessment results using scoring rubric (Table 1).

**Table 1 ijerph-18-02990-t001:** Scoring rubric used to evaluate products relative to each of the six assessment attributes.

Attribute	Level	1	2	3	4	5
	Generic Rubric Scale	No indication will meet our needs	Likely will meet our needs with significant engineering	Meets our needs in a suboptimal way	Meets our needs but has not been validated	Validated and published in peer-reviewed journals
I	Chest Motion Data	Not available	Only TC	TC and AB Sample rate ≤ 100 HZ	TC and AB Sample rate ≥ 100 HZ	Level 4 plus validated and published
II	Data Analysis Software	Signal Conditioning	SDK or API	Calibration	Inhalation topography	Level 4 plus validated and published
III	Ambulatory Observation Capabilities	No information	In-lab or less than a full day in NE	Up to 24 hours uninterrupted in NE	Suitable for multiday with in-field charging or data off-loading	Level 4 plus validated and published
IV	Participant Acceptability	Not acceptable	Indiscreet and cumbersome	Indiscreet or cumbersome	Discreet and not cumbersome	Level 4 plus validated and published
V	Purchasing Constraints	No longer available	Academic research lab	Restricted access to purchase	Available through vender	Available from the vendor’s store front
VI	Affordability	No information	High cost and multiple garments sizes/genders needed	Low cost, but multiple garments sizes/genders needed	High cost, unisex, fewer sizes needed	Low cost, unisex, fewer sizes needed

**Table 2 ijerph-18-02990-t002:** Product characteristics grouped by assessment attributes for each Wearable Respiratory Monitor. NA = Not Applicable, NR = Not Reported.

Brand	LifeShirt	Equivital	Smartex	Hexoskin	Zephyr	Nox	BioRadio
Model		EQ02 LifeMonitor	WWS	Smart Garment	BioHarness	T3 & A1	
Vendor	Vivometrics Inc.	Hidalgo Inc.	Smartex s.r.l.	Carré Technologies Inc.	Zephyr Technology Corp.	Nox Medical Global	Great Lakes NeuroTechnologies
***(I) Chest Motion Data***							
Sensor type	RIP	Strain gauge	Piezoresistive	RIP	Capacitive	RIP	RIP
Sampling rate	50 Hz	25.6 Hz	25 Hz	128 Hz	25 Hz	100–2000 Hz	250–4000 Hz
Sensor locations	AB & TC	TC	TC	AB & TC	TC	AB, TC, both	AB, TC, both
Raw data available	Yes	Yes	Yes	Yes	Yes	Yes	Yes
***(II) Data Analysis Software***							
Signal conditioning	Yes	Yes	No	No	Yes	Yes	NR
SDK or API support	NR	Yes	NR	Yes	NR	NR	Yes
Calibration	Yes	3rd party	3rd party	3rd party	NR	NR	3rd party
Inhalation topography	No	No	No	No	No	No	No
***(III) Ambulatory Observation***							
Storage capacity	NR	1200 h	400 h	600 h	480 h	12–36 h	90 h
Battery life	NR	48 h	30 h	12–30 h	24 h	14–36 h	8 h
***(IV) Participant Acceptability***							
Garment style	Shirt	Strap	Shirt	Shirt	Strap	Belt	Belt
Worn relative to own clothing	under, over, alone	under	under, alone	under, alone	under	under, over	under, over
Datalogger attachment	Indiscreet	Discreet	Discreet	Discreet	Discreet	Indiscreet	Indiscreet
***(V) Purchasing Constraint***							
Available	No	Yes	Yes	Yes	Yes	Yes	Yes
Unrestricted access in US	NA	Yes	Yes	Yes	Yes	Yes	Yes
Store Front	NA	No	No	Yes	No	No	No
***(VI) Affordability***							
Garment cost	NR	$249	$488	$169–$249	NR	$144 for 20 pairs	$119–$349
Datalogger cost	NR	NR	Included	$230–330	NR	$4000–$14,000	$3490
Garment gender options	Unisex	Unisex	F, M	F, M	Unisex	Unisex	Unisex
Garment size ranges	NR	Many	Single	Many	Adjustable	Adjustable	Adjustable

## Data Availability

The data presented are contained within the article.

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
