# Peer review of "Feasibility Assessment of Wearable Respiratory Monitors for Ambulatory Inhalation Topography"

_ijerph, 2021, doi:10.3390/ijerph18062990_

Round 1

Reviewer 1 Report

In this manuscript, the authors reviewed the feasibility assessment of wearable respiratory monitors for ambulatory inhalation topography. I think that the manuscript may be considered as potential publication in Journal of International Journal of Environmental Research and Public Health after addressing these issues as follow.

1. There are too few figures in this article. It is suggested to add some important figures.

2. The manuscript lacks an outlook on future trends of wearable respiratory monitors for ambulatory inhalation topography.

Reviewer 2 Report

Jayasekera et al. evaluated the feasibility of adapting commercially available Wearable Respiratory Monitors (WRM) for measuring smoking inhalation topography. The topic is interesting and the manuscript is well written. My concerns are:

1-    A detailed description of the parameters to be computed from lung volume changes in time and their required accuracy is missing. This is needed to understand the required quality of the measured lung volume signal and therefore, the technology that can be used.

Line 159. It is not clear why a so high sampling frequency is needed.

2-    If smokers' compensating mechanisms modify the abdomen and chest-wall relative contributions to the total lung volume is another critical point to consider. In this case, measuring only the thoracic movements result in measurement errors that are not identifiable after the recording.

3-    As no WRM is already suited for measuring smoking inhalation topography. The evaluation of the difficulties to make the device usable to this aim may be of interest, considering that software improvements are more manageable than changes in the device's hardware.

4-    The ability to accurately sense chest wall motion (and therefore, lung volume) is the essential requirement without that any other desirable characteristic is unuseful. However, due to the lack of information on this topic, it is not considered in the evaluation of the devices. This data are very difficult to obtain as the accuracy is influenced by the device (accuracy and stability of the calibration coefficients over time), the performed calibration, changes in the subject breathing patterns, displacement of the sensors, body position, etc. However, at least some general qualitative consideration may be added.

Line 163-166. Sensor type may play a role, but also its location, the amount of the chest wall it is sensing, and its calibration accuracy are to be considered. For example, strain gauge sensors detect changes of the chest wall portion they cover: the smaller it is, less unlike to represent changes in lung volume accurately.

Line 197. The sampling frequency seems the least critical factor for me. Accuracy and stability of calibration parameters are of first importance. Proper procedures for calibration and calibration repetition must be defined before WRM can be used for measuring smoking inhalation topography.

Line 458-461. I think the sentence written in this way may be questionable. Even if motion artefacts and other noises need to be identified and removed, automatic identification of tidal breathing and breathing pattern parameter computation is much easier than computing accurate lung volume changes from chest wall movements. However, the automatic detection of smoking and non-smoking breaths poses several challenges and may require the incorporation of the hand gesture.

Line 494-496. It is not clear to me why the absence of motion cannot be interpreted as breath-hold.

Reviewer 3 Report

The authors tried to present a feasibility assessment of wearable respiratory monitors for ambulatory inhalation topography to the readers. The research topic is interesting and worth investigating. However, the paper covers 9 WRMs only and the authors tried to categorize such report as a review paper. I do not think it appropriate to do so.

On the other hand, the authors' contribution is questionable and limited: "the results indicate that there are WRMs with core technologies and characteristics that can be built upon for ambulatory topography measurement in the NE." To my understanding, this could be counted as neither theoretical nor technical contribution.

The authors are strongly suggested adding more figures/photos. These could better help the readers understand the devices' working mechanisms. If you do not make figures yourselves, go ahead and ask for the permission of reproducing the photos of those products. The current version is more like a technical report than a review manuscript to be accepted for publication in a journal. 

Round 2

Reviewer 3 Report

Thanks for addressing most of my concerns in the first round. The paper is in better shape now.